# Possible Effects of Early Maternal Separation on the Gut Microbiota of Captive Adult Giant Pandas

**DOI:** 10.3390/ani12192587

**Published:** 2022-09-27

**Authors:** Xiaohui Zhang, Xueying Wang, James Ayala, Yuliang Liu, Junhui An, Donghui Wang, Zhigang Cai, Rong Hou, Mingyue Zhang

**Affiliations:** 1Chengdu Research Base of Giant Panda Breeding, Chengdu 610081, China; 2Sichuan Key Laboratory of Conservation Biology for Endangered Wildlife, Chengdu 610081, China; 3Sichuan Academy of Giant Panda, Chengdu 610081, China

**Keywords:** captive giant panda, maternal separation, stress, gut microbiota

## Abstract

**Simple Summary:**

In the process of ex-situ conservation, young giant pandas face a variety of unfavorable environmental impacts such as frequent maternal separation, parenting by non-parents, noise from tourists, and frequent replacement of animal houses, which may cause psychological and physiological stress. The gut microbiota is an important carrier of the interaction between the body and the environment, and recent studies revealed an association between stress and alterations of the intestinal microbiota. So, is the stress caused by the unfavorable parenting environment in the early life of captive giant pandas related to the gut microbiota? To answer this question, we use fecal metagenomics and LC-MS technology to study the effect of different parenting patterns on the structure, diversity, and metabolites of the intestinal microbial community of captive giant pandas. In order to evaluate the possible adverse effects of the traditional parenting mode on the gut microbiota of captive giant pandas in the early life, it can provide an important scientific basis for improving the welfare level of captive giant pandas.

**Abstract:**

Maternal deprivation (MD) in early life induces dysbiosis in the host gut microbiota, which is a key determinant of abnormal behavior in stress model individuals. Compared with the early parenting environment of the wild, captive giant pandas face frequent and premature maternal separation. Will this lead to imbalance in intestinal flora and stress in captive giant pandas? The purpose of this research is to evaluate the possible adverse effects of the traditional parenting mode on the gut microbiota of captive giant pandas. The results showed that the frequent and premature maternal separation at early stages of the young did not change α and β diversity indices of the gut microbes, but it increased the relative abundance of s_*Clostridium_tetani* and s_*Clostridium_sp_*MSJ_8 (significantly positively correlated with the metabolism of propionic acid) and also the concentrations of fecal metabolites that are related to stress (N-acetyl-l-aspartic acid and corticosterone) in the intestinal tract of giant pandas in adulthood. Thereby, the function of protein digestion and absorption in the intestines of captive giant pandas was decreased, and the metabolism of short-chain fatty acids was disturbed. In conclusion, the parenting experience of early maternal separation could adversely affect the stress caused by the unfavorable parenting environment in the early life of captive giant pandas related to the gut microbiota of the captive giant pandas in adulthood.

## 1. Introduction

The harsh environment in early life is one of the important factors affecting the long-term health of humans and animals. For cubs, maternal deprivation (MD) is the most common early life harsh environment that has a profound impact on adult behavior and development and can produce significant stress responses [1]. When individuals are subjected to short-term stress or continuous chronic stress, they often show abnormal emotional and behavioral abnormalities [2]. In the occurrence and development of various psychological and physiological diseases caused by chronic stress, the disturbance of intestinal microbial metabolism is one of the most important factors [3]. The gut microbiome is an important carrier for the interaction between the body and the environment. The neural network system of the gut is also known as the “second brain”. Signals from the gastrointestinal tract can be transmitted to the brain through the brain-gut-microbiome axis, triggering the body’s psychological stress and physiological responses [4]. The brain-gut-microbiome axis is an important pathway for the connection between the brain and the gut [3], and stress increases gut permeability, resulting in long-term changes in gut microbial diversity and abundance [2,5]. The gut microbiota uses pathways such as intestinal short-chain fatty acid and tryptophan metabolism to realize the process of information exchange between the brain and the gut. Short-chain fatty acids (SCFAs) can cross the blood-brain barrier to reach the brain and indirectly send signals to the brain through neural activation, thereby affecting physiology and behavior [4].

A close follower-type relationship between mother and cub early in life (nursing) is extremely important for the behavioral, psychological, and physical development of wild giant panda cubs [6]. In the process of ex-situ conservation, young giant pandas face a variety of unfavorable environmental impacts such as frequent maternal separation, parenting by non-parents, noise from tourists, and frequent replacement of animal houses, which may cause psychological and physiological stress. This causes stress to the mothers during the nursing period [7], resulting in less opportunities for learning and exercise when rearing the cubs, and also affects the social behavior of giant panda cubs to a certain extent [8]. Since the cubs are weak at the young stage, to increase the survival rate the traditional parenting mode usually adopts artificial milk feeding apart from the breast feeding by mothers. Recent studies revealed an association between stress and alterations of the intestinal microbiota [9] and specific gut bacteria restrain the activation of the hypothalamic–pituitary–adrenal (HPA) axis and affect social behaviors [9,10]. Therefore, we proposed a conjecture, that is, whether parenting experience will change the structure of the intestinal flora of captive giant pandas. The diversity of the human gut microbiome was initially described using 16S ribosomal RNA gene sequencing [11], and later shotgun sequencing showed that the gut microbiome is closely related to host physiology, health, and disease [12]; therefore, these methods have been widely used to explore the adaptation and evolution of gut microbes in wild animals.

Evaluating the possible adverse effects of the traditional parenting mode on the gut microbiota of captive giant pandas can provide an important scientific basis for improving the welfare level of captive giant pandas. Therefore, the present research uses non-invasively fecal metagenomics and liquid chromatography mass spectrometer (LC-MS) technology to study the effect of different parenting pattern on the structure and diversity of the intestinal microbial community and metabolites of captive giant pandas.

## 2. Materials and Methods

### 2.1. Feces Samples Collection

The feces collection of captive giant pandas in this experiment was carried out at the Chengdu Research Base of Giant Panda Breeding. A total of 12 healthy adult giant pandas were divided into two groups (Adult Parent-Raised group and Adult Hand-Raised); a total of 12 fecal samples of captive giant pandas were collected (one from each panda) (Table 1). Fecal samples were collected within ten minutes after the giant panda defecates, and the outer layer of the feces in contact with the ground was stripped, packed into sterile bags and air expelled, put in an ice box, transported back to the laboratory, and stored at −80 °C.

### 2.2. Feeding and Management

The developmental stages of giant pandas are mainly divided into the juvenile stage, sub-adult stage, and adult stage [13]. Giant pandas from birth to 2 years old before leaving their mothers are collectively referred to as the juvenile stage; around 2 years old, giant pandas enter the sub-adult stage from the juvenile stage and live independently from their mothers; and in captivity, females attain maturity at 4.5 years and males at around 6 years. The present study categorized the parenting methods into two: individuals exclusively breastfed and nursed by parents as Parent Raised (PR) and those artificially assisted by a human as Human Raised (HR). The exclusive breastfeeding nursing method is divided into two types: mother raising singleton and mother raising twins. In the early stage of parenting (the first 3 months), the singleton cubs are breastfed throughout the whole process and do not need supplemental milk. The twin cubs are breastfed in turn and need to be supplemented with formula milk during this period, and the singleton cubs spend more time with their mothers than the twins; the artificial assisted nursing method is mainly aimed at the mothers with poor maternity or insufficient breast milk that cannot raise their own single or twin cubs by themselves. All these cubs were raised by artificial collection of colostrum and formula milk, and all twin cubs also adopted this feeding method and were raised in incubators.

All giant panda cubs are weaned at the age of 1.5. After weaning, all adopt the method of gradual separation of mother and child to transition to sub-adult feeding and management, so as to reduce the stress caused by separation of cubs and their mothers. During the period, a small amount of bamboo leaves and shoots were provided for the juveniles gradually to switchover from milk feeding to fodder feeding. At the age of 2 years onwards, all the cubs in the present study are independent of mothers. The feeding and management mode and food supply of sub-adults are completely the same. Generally, 3–5 giant pandas born in the same year share the outdoor sports field during the day and rest alone at night. Therefore, the diet types of giant pandas adopted different nursing methods are the same during the rearing stage (nursing stage, sub-adult stage, and adult stage). For age 0–8 months old giant pandas only consume milk as their diet (breast milk and formula), 9–18 months giant pandas consume old milk (breast milk and formula) and bamboo as their diet, and after 18 months old, giant pandas only consume bamboo as their diet.

### 2.3. Total DNA Extraction and Library Construction

Due to the large amount of undigested bamboo in giant panda feces, pretreatment is required before total DNA extraction. Take about 10 g of feces into a 10 mL centrifuge tube, add 8 mL of sterilized PBS, and vortex for 5 min to remove the microorganisms in the feces from the particles in the feces such as bamboo, and collect the supernatant. After collecting the supernatant three times, centrifuge at 12,000 rpm for 10 min to collect the microbial pellet and store at −70 °C. Precipitated total DNA was extracted using the Omega Fecal DNA Extraction Kit, the purity and integrity of the extracted DNA were analyzed by 1.5% agarose gel electrophoresis, and the quality and quantity of the DNA were detected by the SmartSpec plus nucleic acid protein analyzer to ensure that the concentration and purity of the DNA met the requirements (OD_260_/OD_280_ value between 1.8 and 2.0).

Qualified DNA samples were randomly broken into fragments of about 350 bp in length with ultrasonic liquid processors (Covaris company, Woburn, MA, USA), and the entire library was prepared through the steps of end repair, A-tail addition, sequencing adapter addition, purification, and PCR amplification. After the library was constructed, we used Qubit2.0 for preliminary quantification, diluted the library to 2 ng/ul, and then used Agilent 2100 to detect the insert size of the library. After the insert size met expectations, we used the Q-PCR method to determine the effective concentration of the library to ensure library quality. After the library inspection was qualified, the different libraries were pooled according to the requirements of effective concentration and targeted data volume [14].

### 2.4. Metagenomic Shotgun Sequencing of Microbialcommunities

The prepared library was sequenced on the Illumina PE150 sequencing platform, followed by double-ended fill-in sequencing (2 × 150 bp). This step was completed in Shanghai Applied Protein Technology Co., Ltd. After obtaining the sequencing data, first use the FASTX-Toolkit (Version 0.0.13) (http://hannonlab.cshl.edu/fastx-toolkit, accessed on 19 August 2022) to filter the data (clean), including: 1. Remove the reads containing two Ns. 2. Remove the adapter in the sequence according to the primer information. 3. Cut off the reads with low quality value, and finally process the results obtaining valid sequence (clean reads). Next, the high-quality sequences are corrected and assembled to construct a metagenomics contigs sequence set. After de-redundancy, gene prediction is performed to obtain a non-redundant amino acid sequence set. After preprocessing, Clean Data was obtained, and then SOAP denovo assembly software was used for assembly analysis. For a single sample, first we selected different K-mer = 55 for assembly, and then selected the Scaffolds with the largest N50 as the final assembly result of the sample. We interrupted the assembled Scaffolds from the N coFnnection to obtain a sequence fragment without N (called scaftig), and then counted all Scaftigs data in the assembly result. The assembled contigs were then aligned to MetaPhlAn2 [15] (http://segatalab.cibio.unitn.it/tools/metaphlan2/, accessed on 19 August 2022) and MBGD (Microbial Genome) using blastn database (http://mbgd.nibb.ac.jp/, accessed on 19 August 2022) [16,17] to obtain species-level annotation information, then use the sum of the gene abundances corresponding to the species to calculate the abundance of the species, obtain the species composition spectrum at the fine level of species and below, and perform species composition analysis (based on MEGAN, GraPhlAn’s taxonomic composition visualization, and Krona’s taxonomic composition interactive display), α diversity and β diversity analysis (As data of the Chao1 are normally distributed, analyzed by T-test, and the data of the Simpson and Shannon are not normally distributed, tested using Wilcox Matched Pair Test), difference analysis (species Venn diagram, linear discriminant analysis (LDA), effect size analysis, partial least squares discriminant analysis (PLS-DA), random forest analysis, etc.).

### 2.5. LC-MS/MS Analysis

The frozen fecal samples were slowly thawed at 4 °C, an appropriate amount of samples was added to pre-cooled methanol/acetonitrile/water solution (2:2:1, *v*/*v*), mixed by vortex, sonicated at low temperature for 30 min, and allowed to stand at −20 °C 10 min, centrifuged at 14,000 rpm for 20 min at 4 °C, taken the supernatant and dry it in vacuo, added 100 μL of acetonitrile aqueous solution (acetonitrile: water = 1:1, *v*/*v*) to reconstitute, vortex, and centrifuged at 14,000 rpm at 4 °C for 15 min, taken the supernatant sample for analysis.

Analysis was performed using an UHPLC (1290 Infinity LC, Agilent Technologies, Santa Clara, CA, USA) coupled to a quadrupole time-of-flight (AB SciexTripleTOF 6600) in Shanghai Applied Protein Technology Co., Ltd (Shanghai, China). For the detailed detection steps, please refer to our previously published paper [18]. The general process includes QC sample preparation, LC-MS, and data analysis. Data analysis includes univariate statistical analysis, multidimensional statistical analysis, differential metabolite screening, differential metabolite correlation analysis and Kyoto encyclopedia of genes and genomes, http://www.kegg.jp/ (accessed on 19 August 2022) (KEGG) pathway analysis.

In order to more comprehensively and intuitively show the relationship between samples and the difference in expression patterns of metabolites such as propionate in different samples, we subtracted the mean value of the group to which all samples and differential metabolites belonged, and divided by the root mean square of this group for normalization, then calculated the distance matrix and used hierarchical clustering for clustering. At the same time, in order to measure the metabolic proximities between significantly different metabolites and further understand the mutual regulation relationship between metabolites in the process of biological state changes, we performed correlation analysis to obtain a correlation heatmap and grid map. KEGG metabolic pathway enrichment analysis was performed on metabolites in feces, and Fisher’s exact test was used to analyze and calculate the significance level of metabolite enrichment in each pathway, so as to identify significantly affected metabolic and signal transduction pathways.

### 2.6. Correlation Analysis

The relative abundances of 7 bacterial groups with significant differences at the species level obtained by metagenomic sequencing analysis in all experimental samples and the expression levels of 32 significantly different metabolites obtained by metabolomics analysis were arranged in a table. The Spearman analysis method was used to calculate the correlation coefficient between the significantly different microbiota and the significantly different metabolites in the experimental sample and draw the correlation matrix heat map.

## 3. Results

### 3.1. Effects of Early Nursery Environment on Gut Microbial Composition of Captive Giant Pandas

From the results in Figure 1a, it can be seen that the intestinal flora of each group of giant pandas are dominated by Proteobacteria, Firmicutes, Actinobacteria, Uroviricota, Bacteroidetes, and Basidiomycota, with the first two phyla accounting for 75% of the total microbial community.

It can be seen from the results in Figure 1b that these 20 microorganisms account for about 50% of the total number of communities. From the results in Figure 1b, it can be seen that the relative abundances of *Escherichia*, *Citrobacter*, and *Enterobacter* under Proteobacteria and *Clostridium* under Firmicutes in the AHR group are lower than that of the APR group, while the relative abundance of *Klebsiella*, *Kluyvera*, *Acinetobacter,* and *Dechloromonas* under *Proteobacteria* and *Lactococcus* under Firmicutes were higher than that of the APR group.

### 3.2. Effects of Early Nursery Environment on Gut Microbial Diversity in Captive Giant Pandas

Table 2 is the rank sum test of simpson, shannon, and chao1 index, showing the alpha diversity results of the APR group and AHR group. It can be seen from Table 2, Appendix A that there is no significant difference in the α-diversity index of the gut biome between the PR and HR groups (simpson, shannon, and chao1 indices: *p* > 0.05) (Table 2).

Figure 2 is a PcoA cluster analysis diagram of gut microbial β diversity in captive giant pandas adopted with different nursing methods. As can be seen from Figure 2, the gut microbial communities of the captive giant pandas in the APR group and AHR group are similar. It can be seen from Table 3 and Table 4 that there is no significant difference in the β-diversity index (Adonis and ANOSIM) of the captive giant pandas’ gut microbes among the groups (*p* > 0.05).

The LefSe analysis histogram shows the microorganisms whose LDA score is greater than the set value (the default value is 2) for the two groups in different species (Figure 3a). Although the differences in β diversity between the two groups were not significant, there are still species with significant differences in relative abundance (*p* < 0.05). For example, in comparing the results of different nursing methods on the gut microbes of captive giant pandas, it is indicated that there are two species with significant differences in abundance in the APR group, *Enterobacter* and *Lelliottia* at the phylum level of Proteobacteria, while the AHR group had five species with significant differences in abundance, namely *Siphoviridae* at the Uroviricota phylum level, and *Bacillus*, *Clostridium*, *Sarcina*, and *Streptococcus* at the phylum level of Firmicutes.

The functional abundance tables of the KEGG and CAZy databases are analyzed by LefSe, respectively. Comparing the effects of different parenting methods on the functional groups of adult giant pandas’ gut microbes, it indicates that there are four types are significantly enriched in the APR group, such as carbon_fixation_pathways_in_prokaryotes; the AHR group has three types are significantly enriched, such as tropane_piperidine_and_pyridine_alkaloid_biosynthesis (Figure 3b). In addition, the CAZy enzyme functional module annotation results show that the largest number of protein families were annotated to the corresponding glycoside hydrolases (GHs) and glycosyltransferases (GTs), which are 56,674 and 75,376, respectively (Figure 3c).

### 3.3. Effects of Early Nursery Environment on Fecal Metabolites in Adult Captive Giant Pandas

The comparison of the total ion chromatogram (TIC) of the QC samples in this experiment and the principal component analysis (PCA) evaluation of the total samples are used to comprehensively evaluate the stability of the instrument, the repeatability of the experiment, and the reliability of the data quality. The results show that the response intensity and retention time of each chromatographic peak basically overlapped, indicating that the variation cause by instrument error during the entire experiment is small (Appendix A). Moreover, the QC samples in the positive and negative ion modes are closely clustered together, indicating that the experiment is reproducible (Appendix A). Therefore, the instrumental analysis system of this experiment is stable, the experimental data is stable and reliable, and the differences in the metabolic profiles obtained in the experiment can reflect the biological differences among the samples.

Based on univariate statistical analysis methods, differential analysis is performed on all metabolites detected in positive and negative ion modes (FC > 1.5 or FC < 0.67, *p*-value < 0.05), and a volcano plot is used for visual display. The significant difference metabolites with qualitative names are selected to mark the up-regulated top 10 and down-regulated top 10 for their expression changes. The results are shown in Appendix A. Based on multi-dimensional statistical analysis method, PCA analysis is performed on all metabolites detected in positive and negative ion modes. The PCA model parameters obtained by 7-fold cross-validation are R^2^X = 0.557 (positive ion mode) and R^2^X = 0.603 (negative ion mode), the results show that the degree of aggregation within the group is high, and the separation between the groups is obvious, indicating that the model is reliable and there are significant differences between the groups. At the same time, the PLS-DA method is used to establish the relationship model between metabolite expression and sample category to realize the prediction of sample category. The PLS-DA model parameters obtained by 7-fold cross-validation are R^2^X = 0.960 (positive ion mode) and R^2^X = 0.961 (negative ion mode), and the results show that the model has good stability (Appendix A). In order to avoid overfitting of the supervised model in the modeling process, the permutation test is used to test the model to ensure the validity of the model. The results show that with the gradual decrease of the permutation retention, the R2 and Q2 of the stochastic model gradually decrease, indicating that the original model did not have overfitting and the model was robust (Appendix A).

In this experiment, strict OPLS-DA VIP > 1 and *p*-value < 0.05 are used as the screening criteria for significant differential metabolites, and a total of 32 differential metabolites are screened (as shown in Table 5). Among them, compared with the AHR group, the down-regulated metabolites in the APR group are 13 kinds, which mainly include guanidinopropionic acid, alpha-guanidinoglutaric acid, calpain inhibitor I, etc., and the up-regulated metabolites are 19 kinds, which mainly include propionic acid, 5 -aminolevulinic acid, methylmalonic acid, etc., mainly belonging to lipids and lipid-like molecules, organic acids and derivatives, organic nitrogen compounds, and other metabolite categories.

Hierarchical clustering of each group of samples is performed using the expression of metabolites with qualitatively significant differences. The results show that the fecal metabolic characteristics of adult captive giant pandas changed significantly due to the parenting method, and the changing trends of these metabolites are described by a heat map (Figure 4a); metabolite correlation analysis show that these metabolites interacted (Figure 4b), including a positive correlation between propionic acid and methylmalonic acid. Correlation analysis is performed between the screened microorganisms with differential levels and differential metabolites. The results show that s_*Clostridium_tetani* and s_*Clostridium_sp_*(MSJ_8) are significantly positive correlated with propionic acid (r = 0.46, *p* < 0.05; r = 0.61, *p* < 0.01) and methylmalonic acid (r = 0.38 *p* < 0.05; r = 0.39, *p* < 0.05) related to propionic acid metabolism; s_*Lelliottia_amnigena* is significantly negatively correlated with N-acetyl-l-aspartic acid (r = −0.52, *p* < 0.05); and s_Enterobacter_sp_638 is significantly negatively correlated with corticosterone (r = −0.46, *p* < 0.05)(Figure 4b). It can be seen from the KEGG pathway analysis results that there are four major metabolic pathways with significant differences, namely: protein digestion and absorption, propionate metabolism, regulation of lipolysis in adipocytes, and ethylbenzene degradation (Figure 4c).

## 4. Discussion

Premature and frequent maternal separation is the most prevalent stressful environment in early life and is a determinant of various diseases including brain and gut dysfunction [19]. Artificially assisted nursing method is a traditional management mode of captive giant pandas during the nursing period, which leads to frequent maternal separation in the early nursing period. Our previous research found that the artificially assisted nursing method might increase the frequency of stereotypic behaviors and the levels of urinary cortisol in adult female giant pandas in captivity [7]. These all seem to indicate that the artificially assisted nursing method of maternal separation in the early parenting stage have produced stress on captive giant pandas, but is this related to the stress caused by the unfavorable parenting environment in the early life of captive giant pandas related to the changes of gut microbiota caused by the nursing environment? The study of the intestinal flora of Siberian tigers, which are also endangered wild animals, found that the living environment could significantly change the alpha diversity of the intestinal flora of Siberian tigers, resulting in significant differences in gut microbiota composition and function between captive and wild Siberian tigers populations [20]. Likewise, the relative abundances of the gut microbiota between captive and wild forest musk deer were significantly different [21]. Therefore, this experiment deeply studied the effect of early parenting experience on the intestinal microbial diversity and intestinal bacterial metabolites of captive giant pandas. The results showed that the parenting mode did not have a significant impact on α and β diversity indices of the intestinal flora of captive giant pandas, but it significantly affected the abundance of some specific intestinal flora and the expression of intestinal bacterial metabolites. This study compared the gut microbiota structure of captive giant pandas, and the results indicated that the gut microbiota of captive giant pandas was mainly composed of Firmicutes and Proteobacteria, which was also consistent with previous studies [22,23], but there were significant differences in the proportions of these two major phyla between age groups and between groups of different nursing methods, which might be due to the different physiological states of the hosts themselves. The present study found that the relative abundance of *Clostridium* and two species under the genus s_*Clostridium_tetani* and s_*Clostridium_sp_*MSJ_8 in the APR group was higher than that in the AHR group. As an important genus of SCFAs produced by intestinal fermentation, *Clostridium* was involved in the synthesis of various SCFAs [24]. Under the action of anaerobic microorganisms in the large intestine of mammals, carbohydrates (CHO) were degraded with the help of a variety of bacteria and fermented to produce a large number of SCFAs, including acetate, propionate, and butyrate, which played important roles in the regulation of host energy metabolism, gut homeostasis, and host immune system [25]. In order to verify our guess, we then performed LC-MC/MC analysis; the results also indicated that the propionate metabolism related to the synthesis of SCFAs (propionic acid) and protein digestion and absorption pathway was significantly enriched, and the related metabolites propionic acid, methylmalonic acid, and N-acetyl-l-aspartic acid were also significantly increased in the APR group. The results of spearman correlation analysis also indicated that s_*Clostridium_tetani* and s_*Clostridium_sp_*MSJ_8 were significantly positively correlated with propionic acid and methylmalonic acid related to propionic acid metabolism. At the same time, the CAZy enzyme functional module annotation results also found that the number of protein families corresponding to glycoside hydrolases (GHs) and glycosyltransferases (GTs) was the largest, which also reflected that the parenting experience did indeed affect the protein digestion and absorption capacity of the captive giant panda’s intestine. N-acetyl-l-aspartic acid, as an important marker reflecting the function of the nervous system [24,26,27], was associated with a variety of central nervous system diseases (depression) [26]. However, our results also showed that the concentration of the metabolite corticosterone in the feces of the APR group was significantly higher than that of the AHP group. As an important physiological indicator reflecting stress, the increase in the concentration of corticosterone indicated that there was stress in the animal body [28], but there had also been studies showing that acute corticosterone elevations were thought to be beneficial for physical health compared to the deleterious effects of chronic stress conditions [29]. At the same time, we should also be aware that the stool samples collected in this experiment were from adult giant pandas. Many years had passed since the maternal separation occurred. Although later the management and feeding conditions were the same for the two groups, there might also be other factors, such as age (mean age was higher in the AHR group), that could have influenced the differences. These results all indicated that environmental stress in the early nursing period (frequent and premature maternal separation) would have a negative impact on the psychology of captive giant pandas in adulthood. Our previous study showed that parenting environment could significantly increase the expression frequency of abnormal behavior and the concentration of urinary cortisol [7]. From the perspective of gut microbes, the results of this research once again proved that the parenting experience of early maternal separation could cause stress in giant pandas and adversely affect gut microbiota in adulthood.

## 5. Conclusions

In conclusion, our results showed that the maternal separation environment in the early parenting period did not affect the species diversity of the gut microbiota of captive giant pandas, but it had a significant impact on the abundance of microbes related to propionic acid metabolism and the concentration of stress-related fecal metabolites in the intestinal tract of captive giant pandas after adulthood. As a result, the function of protein digestion and absorption in the intestines of captive giant pandas was decreased, as well as the metabolic disorders of SCFAs, which could adversely affect the gut microbiota of captive giant pandas.

## Figures and Tables

**Figure 1 animals-12-02587-f001:**
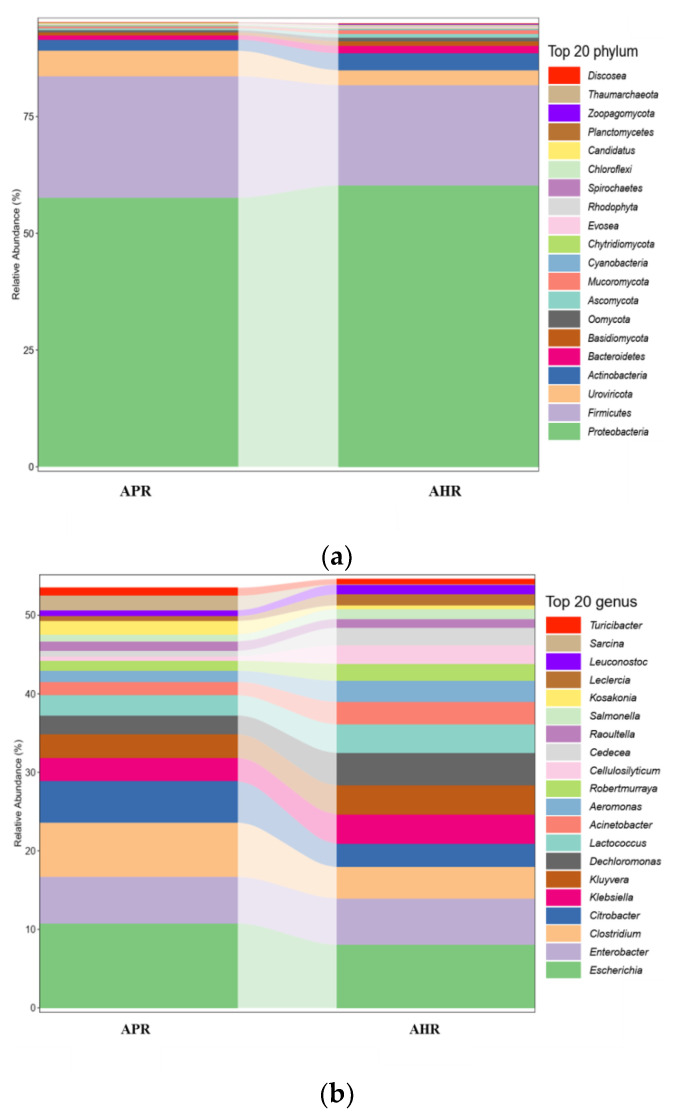
Column chart of relative abundance at the phylum (**a**) and genus levels (**b**) in fecal microbiota of captive giant pandas adopting different nursing methods. Note: In the figure, the abscissa is arranged according to the sample name, each bar represents a sample, and each taxon is distinguished by color, and the ordinate represents the relative abundance of each taxon, the longer the bar, the higher the relative abundance.

**Figure 2 animals-12-02587-f002:**
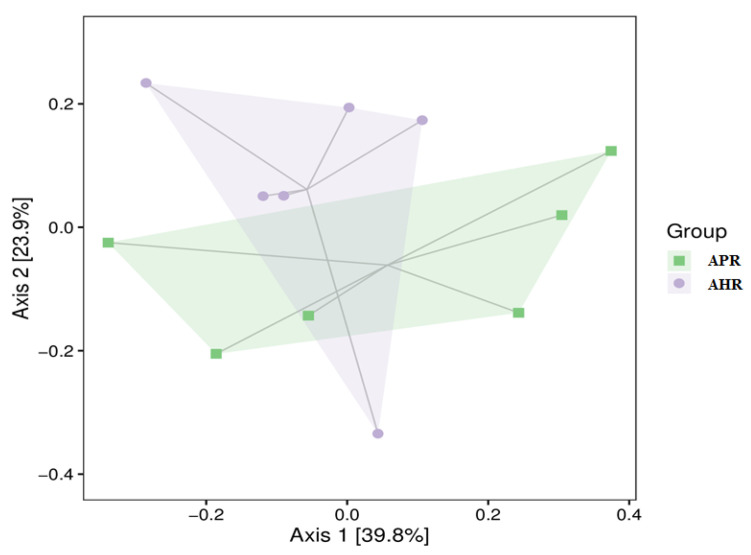
PcoA clustering circle plot in fecal microbiota of captive giant pandas adopting different nursing methods. Note: In the figure, each point represents a sample, and the points with different colors belong to different samples (groups). The closer the distance between the two points, the smaller the difference in the species composition of the two samples and the higher the similarity. The percentages in brackets on the axes represent the proportion of the variance in the raw data that can be explained by the corresponding principal coordinates.

**Figure 3 animals-12-02587-f003:**
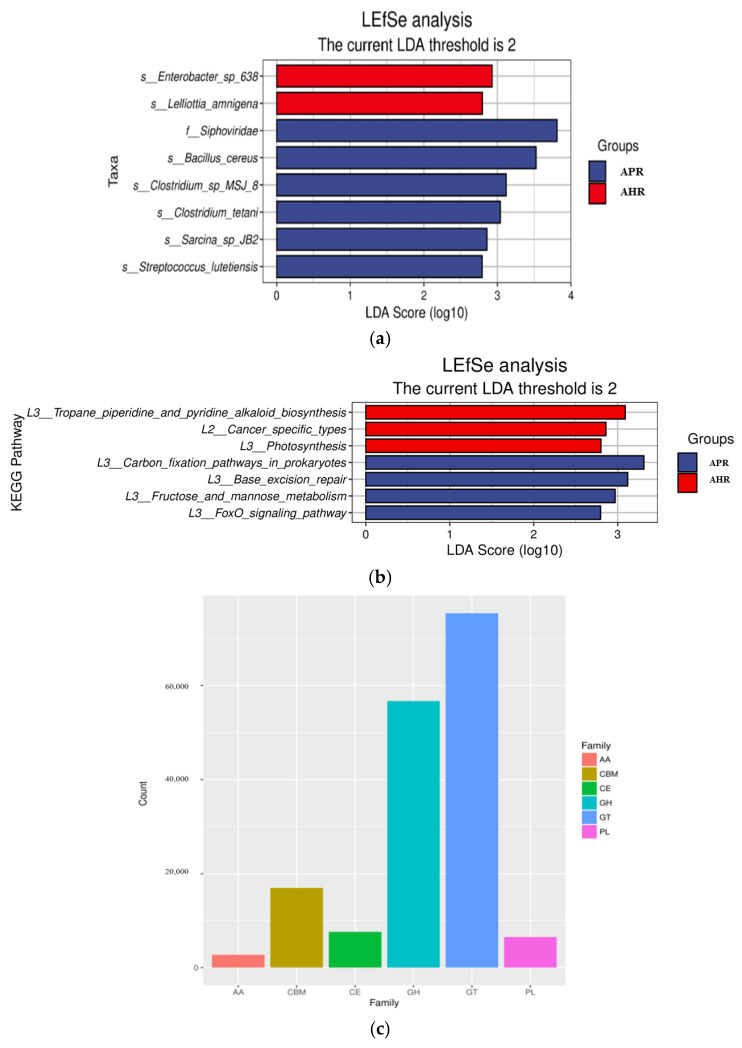
Column chart of LEfSe analysis of gut microbes of captive giant pandas. (**a**) Column chart of LEfSe analysis of gut microbes of captive giant pandas. (**b**) LEfSe functional histogram. Notes: In the figure, the ordinate is the taxa with significant differences between groups, and the abscissa is a bar chart to visually display the LDA analysis logarithmic score value of each taxon. The taxa are ordered by the size of the score value to describe their specificity in the sample grouping. The longer the length, the more significant the difference of the taxon is, and the color of the bar chart indicates the most abundant sample grouping corresponding to the taxon. (**c**) Statistical chart of EggNOG functional group annotation results. Notes: In the figure, the abscissa corresponds to each CAZy enzyme functional module, and the ordinate is the number of protein families annotated to the corresponding module.

**Figure 4 animals-12-02587-f004:**
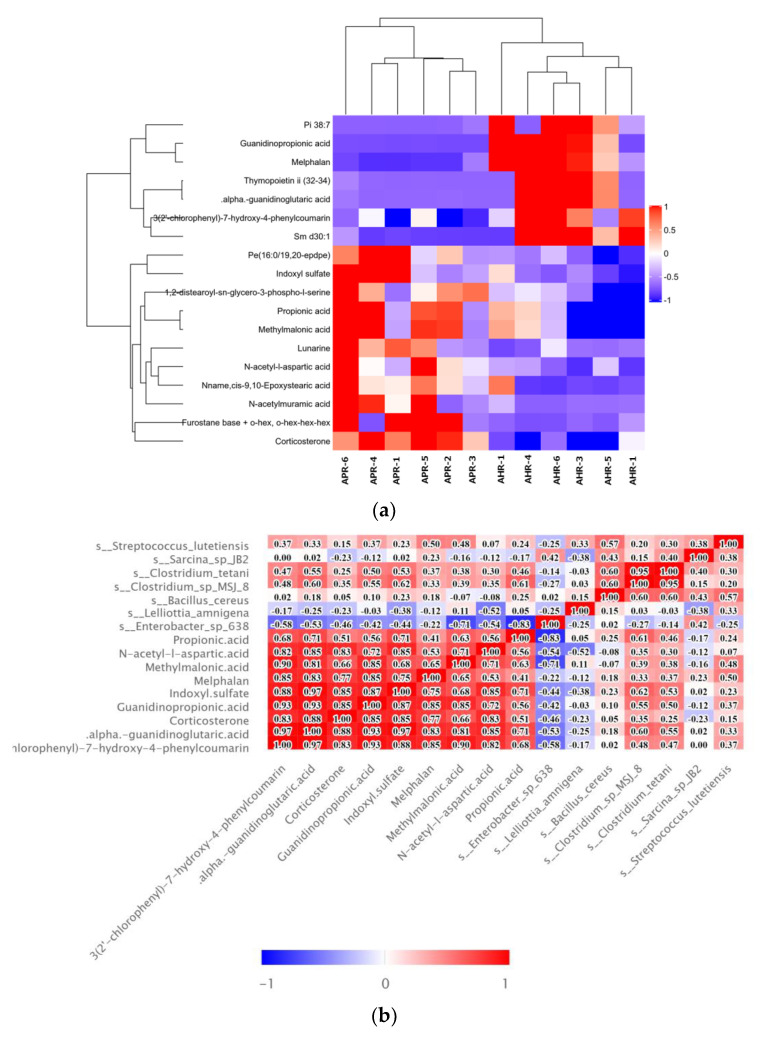
Bioinformatics analysis of differential metabolites. (**a**) Hierarchical clustering heatmap of significant differential metabolites. Note: Each row in the figure represented a differential metabolite, and each column represented a group of samples. Red represented significant up-regulation, blue represented significant down-regulation, and the depth of the color represented the degree of up-regulation. Metabolites with similar expression patterns gather under the same cluster on the left. (**b**) Correlation heatmap of significant differential metabolites and intestinal microorganisms. Note: Red indicated a positive correlation, blue indicated a negative correlation, and white indicated a non-significant correlation. The color depth was related to theabsolute value of the correlation coefficient that was, the higher the degree of positive or negative correlation, and the darker the color was. The size of the square was related to the significance of the correlation, the smaller the p-value, the stronger the correlation, and the larger the square size was. (**c**) KEGG pathway annotation and analysis. Note: The vertical axis in the bar graph represented each KEGG metabolic pathway, and the horizontal axis represented the number of differentially expressed metabolites contained in each KEGG metabolic pathway. The color represented the *p*-value of the enrichment analysis, the darker the color, the smaller the *p*-value was, and the more significant the degree of enrichment, and the number on the column represented the rich factor.

**Table 1 animals-12-02587-t001:** Experimental Grouping.

Groups	Name	Gender	Age	Single or Twins
Adult Hand-Raised (AHR)	Gong Zai	Male	13 years	single
Adult Hand-Raised (AHR)	Ying Ying	Male	13 years	single
Adult Hand-Raised (AHR)	Cheng Shuang	Male	9 years	twins
Adult Hand-Raised (AHR)	Ni Da	Female	6 years	twins
Adult Hand-Raised (AHR)	Cheng Da	Female	10 years	single
Adult Hand-Raised (AHR)	Mei Lun	Female	8 years	twins
Adult Parent-Raised (APR)	Mei Lan	Female	15 years	single
Adult Parent-Raised (APR)	Lou ABao	Male	14 years	single
Adult Parent-Raised (APR)	Mei ABao	Female	11 years	single
Adult Parent-Raised (APR)	ZhiZhi	Female	12 years	single
Adult Parent-Raised (APR)	YaZai	Female	15 years	single
Adult Parent-Raised (APR)	Zhao Mei	Female	11 years	single

Note: All twins in the APR group were fed by mixed feeding of artificial formula milk and breast milk, and all singletons were exclusively breastfed. All the nursing stage in the AHR group was mixed feeding with artificial formula milk and breast milk.

**Table 2 animals-12-02587-t002:** α diversity index of gut microbes in captive giant pandas adopting different nursing methods.

Sample	Simpson	Chao1	Shannon
APR	0.9654 ± 0.0147	4683.64 ± 969.04	6.8373 ± 0.7607
AHR	0.9761 ± 0.0067	5835.38 ± 734.65	7.3393 ± 0.3735

**Table 3 animals-12-02587-t003:** Species composition Adonis difference analysis results between groups.

Items	Df	Sums of Sqs	Mean Sqs	F. Model	R2	Pr(>F)
Treat2	1	0.12481	0.12481	0.97122	0.08852	0.412
Residuals	10	1.28509	0.12851		0.91148	
Total	11	1.40990			1.00000	

**Table 4 animals-12-02587-t004:** Species composition ANOSIM difference analysis results between groups.

Method Name	R Statistic	*p*-Value	Number of Permutations
ANOSIM	0.0722	0.256	999

**Table 5 animals-12-02587-t005:** Difference metabolites identified by positive and negative ion mode.

ID	Adduct	Name	VIP	Fold Change	*p*-Value	m/z	rt(s)
M173T275	[M+H]+	Arcaine	15.98198784	6.63298872	0.007585569	173.13961	275.432
M406T344	[M+Na]+	Calpain inhibitor i	1.069037758	0.105522126	0.028039335	406.25683	344.348
M316T296	[M+H+2i]+	Hexaconazole	1.010155018	9.092924318	0.030197868	316.09947	296.492
M431T215	[M+Na]+	3beta,7beta,12beta-trihydroxy-5beta-cholan-24-oic acid	1.302609451	3.620198553	0.031882452	431.27555	215.146
M375T491_2	[M+Na]+	15-ketoprostaglandin f2.alpha.	2.406682636	1.815201502	0.034250987	375.22361	490.8645
M618T37	[M+Na]+	1-palmitoyl-2-oleoyl-sn-glycerol	1.74647584	0.33129875	0.039691999	617.51147	37.1175
M615T38	[M+Na]+	1-palmitoyl-2-linoleoyl-rac-glycerol	1.8091764	0.447178675	0.040387859	615.49474	37.989
M218T271	[M+H]+	L-propionylcarnitine	1.21936108	0.293714647	0.040439675	218.13761	271.0745
M138T211	[M+H]+	Tyramine	2.710329374	4.316237507	0.04166704	138.09027	210.6815
M635T37	[M+NH4]+	1,2-dilinoleoylglycerol	5.638486128	0.171020689	0.042208688	634.53805	36.9215
M563T168_1	[M+H]+	Protoporphyrin ix	2.179714192	1.796078135	0.044486683	563.26432	167.687
M131T324_2	[M+H]+	N-acetylputrescine	4.312577468	2.540127734	0.047607989	131.11717	323.774
M325T376	[M+H-H2O]+	Melibiose	2.887633512	3.193731624	0.048006152	325.11229	375.8195
M345T201	[M-H]-	Corticosterone	1.334636069	3.073188761	0.000006843	345.22681	200.549
M607T40	[2M-H]-	Melphalan	1.621588379	0.046578615	0.000467639	607.14828	40.001
M392T336	[3M-H]-	Guanidinopropionic acid	1.572600471	0.005931119	0.001473817	392.19985	336.089
M706T175	[M+CH3COOH-H]-	Sm d30:1	1.479218847	0.04733475	0.001823034	705.51643	174.804
M879T235	[M-H]-	Pi 38:7	1.404372826	0.013405137	0.01078511	879.49408	234.755
M1082T38	[M-H]-	Furostane base + o-hex, o-hex-hex-hex	1.234234036	15.77960671	0.011283726	1081.52055	38.444
M347T37	[M-H]-	3(2’-chlorophenyl)-7-hydroxy-4-phenylcoumarin	2.42651536	0.438614595	0.011422429	347.04313	37.466
M779T44	[M-H]-	Pe(16:0/19,20-epdpe)	2.281257798	3.036982278	0.013858763	778.51345	43.697
M566T401	[3M-H]-	.alpha.-guanidinoglutaric acid	1.265640564	0.014467482	0.015477295	566.21791	400.6375
M416T444	[M-H]-	Thymopoietin ii (32-34)	1.260199839	0.02135598	0.01605087	416.2132	443.875
M690T288	[M+Cl]-	7-benzyl-11,14-dimethyl-16-(2-methylpropyl)-10,13-di(propan-2-yl)-17-oxa-1,5,8,11,14-pentazabicyclo[17.3.0]docosane-2,6,9,12,15,18-hexone	1.289120759	0.004845138	0.016215125	690.36559	287.696
M774T44	[M-H-NH3]-	1,2-distearoyl-sn-glycero-3-phospho-l-serine	1.217162339	1.999427667	0.026779114	773.53	44.129
M292T298	[M-H]-	N-acetylmuramic acid	1.021977643	9.110003872	0.030490542	292.10291	298.445
M73T372_2	[M-H]-	Propionic acid	1.666044111	1.931336222	0.032458363	73.0297	372.0775
M117T372_2	[M-H]-	Methylmalonic acid	4.081076931	1.994118874	0.036897873	117.01951	371.987
M212T34	[M-H]-	Indoxyl sulfate	2.006047308	3.797263074	0.037033194	212.00198	34.181
M174T378_2	[M-H]-	N-acetyl-l-aspartic acid	2.274041379	4.23154483	0.037634414	174.04059	377.882
M436T132_2	[M-H]-	Lunarine	1.143728194	3.927292997	0.037699018	436.23573	132.21

Note: adduct: represents the adduct ion information of the compound; Name: represents the name of the metabolite; VIP: represents the variable projection importance, the larger the value, the more important; FC: represents the fold of difference; *p*-value: represents the *p*-value of the significance analysis, the smaller the *p*-value, the more significant the difference; m/z: is the mass-to-charge ratio; rt(s): is the retention time of the metabolite on the chromatogram, that is, the peak time, in seconds.

## Data Availability

The data presented in this study are available on request from the corresponding author.

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
