# Peer review of "Possible Effects of Early Maternal Separation on the Gut Microbiota of Captive Adult Giant Pandas"

_animals, 2022, doi:10.3390/ani12192587_

Round 1
Reviewer 1 Report (Previous Reviewer 2)
The authors have addressed my concerns. The additional information had made it much clearer, although there is still some detail in the methods that could be expanded – perhaps in supplemental material if not in the main text. The companies for the extraction, shearing, sequencing and quantification are listed, but not for end repair, purification and amplification. The negative controls for extraction and library prep were not performed and cannot be subsequently added. While the authors provide assurances that contamination is unlikely, given the comparisons are within study, and contamination would likely be across all samples this should not present a significant issues with the conclusions drawn in this paper. A couple of minor points
Line 34-35 and elsewhere: These look like the identifiers from the reference library searched. I think these should be listed as Clostridium tetani and Clostridium sp. (MSJ 8).
Line 47: Delete however.
Line 78: HPA axis – define this – i.e. Hypothalamic–pituitary–adrenal (HPA) axis.
Line 97: 12 fecal samples were collected from 12 pandas? Is that one from each panda?
Author Response
We thank you very much for giving us an opportunity to revise our manuscript, we appreciate editor and reviewers very much for their positive and constructive comments and suggestions on our manuscript entitled “Effects of early maternal separation on the gut health of captive giant pandas”. (ID: animals- 1909422).
Specific responds to the reviewer’s comments:
Questions 1:Line 34-35 and elsewhere: These look like the identifiers from the reference library searched. I think these should be listed as Clostridium tetani and Clostridium sp. (MSJ 8).
Response 1: Considering the reviewer's suggestion, we have changed all the words to “Clostridium tetani and Clostridium sp. (MSJ 8)”.
Questions 2:Line 47: Delete however.
Response 2: Considering the reviewer's suggestion, we have deleted the word of “however”.
Questions 3:Line 78: HPA axis – define this – i.e. Hypothalamic–pituitary–adrenal (HPA) axis.
Response 3: Considering the reviewer's suggestion, we have written the full name of the first occurrence of the Hypothalamic–pituitary–adrenal (HPA) axis in the article.
Questions 4:Line 97: 12 fecal samples were collected from 12 pandas? Is that one from each panda?
Response 4: Yes, the 12 samples collected were indeed one fecal sample from each panda, and we added this to the methods.
Reviewer 2 Report (Previous Reviewer 1)

Author Response
We thank you very much for giving us an opportunity to revise our manuscript, we appreciate editor and reviewers very much for their positive and constructive comments and suggestions on our manuscript entitled “Effects of early maternal separation on the gut health of captive giant pandas”. (ID: animals- 1909422).
Specific responds to the reviewer’s comments:
Question / Comment 1: Against my first comment for the line No.13-16, although the authors mentioned Considering the reviewer's suggestion, we have modify the sentence as “In the process of exsitu conservation, young red pandas face a variety of unfavorable environmental impacts such as premature weaning, frequent maternal separation, parenting by non-parents, noise from tourists and frequent replacement of animal houses, which may cause psychological and physiological stress. However, I don’t see the modification in the revised manuscript, and these sentences [Line No. 13-16] remain with same condition, as how it was in the original version submitted. But I see the suggested revision for the first comment is incorporated in line no 66-69. I suggest again such revision is needed in line no. 13-16 also.
Response 1: Thank you very much for reviewer’s suggestions. Considering the reviewer's suggestion, we have modified the sentence as “In the process of ex-situ conservation, young giant pandas face a variety of unfavorable environmental impacts such as frequent maternal separation, parenting by non-parents, noise from tourists and frequent replacement of animal houses, which may cause psychological and physiological stress” in line 13-16 of the revised manuscript.
Questions 2:Similarly, regarding my earlier Question / Comment 2: for the line No. 18 & 19. Here too the authors mentioned Thank you very much for reviewer’s suggestions, as your suggestion, we have rewritten these sentences as “Recent observations had revealed an association between stress and alterations of the intestinal microbiota [9]. These studies suggested that specific gut bacteria could restrain the activation of the HPA axis, and affected social behaviors [10]” And added two references to prove our point. However, I don’t see any difference in the line 18-19. Please explain.
Response 2: Considering the reviewer's suggestion, we have modified the sentence as “recent studies revealed association between stress and alterations of the intestinal microbiota” in line 17-18 of the revised manuscript.
Questions 3:Line No. 20-32: The purpose of this research is to study the effects of different parenting patterns in the early parenting period on the gut microbial community structure, diversity and fecal metabolites of captive giant pandas.
Not clear, I suggest the authors to rewrite this sentence.
Response 3: Considering the reviewer's suggestion, we have modified the sentence as “The purpose of this research is to evaluate the possible adverse effects of the traditional parenting mode on the intestinal health of captive giant pandas.” in line 28-30 of the revised manuscript.
Questions 4:Similarly, Line No 34-39, not clear, suggest the authors to rewrite. Instead of saying significantly changed, authors can mention increased or decreased. For example, The frequent and premature maternal separation at early stages of the young though not changed α and β diversity indices of the gut microbes, it decreased the relative abundance of s_Clostridium tetani, 34 s-Clostridium sp. MSJ_8. In the present form, it is not clear. Please consider rewriting these sentences.
Response 4: Considering the reviewer's suggestion, we have modified the sentence as “The results showed that the frequent and premature maternal separation at early stages of the young though not changed α and β diversity indices of the gut microbes, it increased the relative abundance of s_Clostridium_tetani,s_Clostridium_sp_(MSJ_8) (significantly positively correlated with the metabolism of propionic acid)and also increased the concentrations of fecal metabolites that related to stress(N-acetyl-l-aspartic acid and corticosterone)in the intestinal tract of giant pandas in adulthood.” in line 30-35 of the revised manuscript.
Questions 5:Line No. 71- 75: Since the new-born cubs of giant pandas are very weak, in order to improve the survival rate of giant panda cubs, the traditional parenting mode usually adopts artificial assistance, that is, manual intervention is adopted in the early parenting stage. The advantage is that it can ensure the safety of the cubs and improve the survival rate of the cubs.
The above four lines could be compressed simply as ‘Since the cubs are weak at young stage, to increase the survival rate the traditional parenting mode usually adopts artificial milk feeding apart from the breast feeding by mothers.
Response 5: Thank you very much for reviewer’s suggestions. Considering the reviewer's suggestion, we have modified the sentence as “Since the cubs are weak at young stage, to increase the survival rate the traditional parenting mode usually adopts artificial milk feeding apart from the breast feeding by mothers.” in line 70-72 of the revised manuscript.
Questions 6:Line No. 76-79: Recent observations had revealed an association between stress and alterations of the intestinal microbiota [9]. These studies suggested that specific gut bacteria could restrain the activation of the HPA axis, and affected social behaviors [10].
Suggest to modify the above sentence into ‘recent studies revealed association between stress and alterations of the intestinal microbiota [9] and specific gut bacteria restrain the activation of the HPA axis and affect social behaviors [9, 10].
Response 6: Thank you very much for reviewer’s suggestions. Considering the reviewer's suggestion, we have modified the sentence as “recent studies revealed association between stress and alterations of the intestinal microbiota [9] and specific gut bacteria restrain the activation of the HPA axis and affect social behaviors [9, 10].” in line 72-75 of the revised manuscript.
Questions 7:Line No. 80-81: thereby affecting the normal expression of their mating behaviors? I think it is outside the focus or not within the focus of the study and if so can be deleted.
Response 7: Considering the reviewer's suggestion, we have deleted this sentence.
Questions 8:Question / Suggestion 11 in my earlier review regarding title of the Table 1: The table title should provide all the details for the reader to understand it clearly and the title should be stand-alone. This suggestion is incorporated.
Response 8: Thanks for your comments.
Questions 9:Line No. The developmental stages of giant pandas are mainly
divided into juvenile stage, sub-adult stage and adult stage. According what reference is these three age-classes? Is there an age-category called cub or not? If so, up to what age, an individual categorized into cub, up to what age juvenile, sub-adult and adult. Please make this statement following standard reference.
Response 9: Considering the reviewer's suggestion, we have cited the classic reference of giant panda ex-situ conservation.
Questions 10:Further. Line No. 111-114: female giant pandas in captivity are about sexual maturity begins at 4.5 years old and male pandas are generally later than females. Sexual maturity occurs at 5.5-6.5 years old, and there are individual differences.
Scientific writing should be crisp but at the same detailed. The above sentence is neither crips nor detailed. The first part of the sentence ‘female giant pandas in captivity are about sexual maturity begins at 4.5 years old’ is not formed properly, second part ‘male pandas are generally later than females’, says males attain maturity later but it does not mention the age in which males mature. The next statement here mentions about ‘Sexual maturity occurs at 5.5-6.5 years old’, but it is not known as sexual maturity of what? and finally a part of the sentence says ‘there are individual differences’ I don’t think it is necessary to mention here the individual differences in maturity, because every one knows that there are individual differences in attaining maturity, as everyone experience it. Therefore, the above sentences can be rewritten something like as follows to be clear for the reader. In captivity, females attain maturity at 4.5 years males at around 6 years.
Response 10: Thank you very much for reviewer’s suggestions. Considering the reviewer's suggestion, we have modified the sentence as “in captivity, females attain maturity at 4.5 years males at around 6 years.” in line 107-108 of the revised manuscript.
Questions 11:Line No. 113-116: From the beginning of estrus, giant pandas enter the adult stage [13]. More than estrus, age should be used to indicate an adult, because there are females which do not come into estrus cycle until their old age that does not mean they are still sub-adults, they are adult yet to start estrus cycle. Therefore, saying from the beginning of the estrus, giant padas enter the adult stage is not standard term.
Response 11: In order not to cause ambiguity, we have deleted these sentences as the reviewer's suggestion.
Questions 12:Line No. 114-116: In the present study, the parenting methods of captive giant pandas are categorized into exclusive breastfeeding nursing method by parents (PR) and artificial assisted nursing method (HR). Instead, the same sentence could be as ‘The present study categorized the ) parenting methods into two: individuals exclusively breastfed and nursed by parents as Parent Raised (PR) and those artificially assisted by human as Human Raised (HR).
Response 12: Considering the reviewer's suggestion, we have modified the sentence as “The present study categorized the parenting methods into two: individuals exclusively breastfed and nursed by parents as Parent Raised (PR) and those artificially assisted by human as Human Raised (HR).” in line 108-111 of the revised manuscript.
Questions 13:Line No. 118-120: In this way, the cubs are mainly raised with their mothers, and the separation of the mother and the cubs is less frequent and the time is short (no more than 2 minutes away from the mother each time). The sentence should be rewritten.
Response 13: In order not to cause ambiguity, we have deleted these sentences as the reviewer's suggestion.
Questions 14:Line No. 126: Authors use two different terms ‘cubs and pups’ Is that right to use these two terms to refer red-panda young one.
Response 14: Considering the reviewer's suggestion, we have changed all the word “pups” to “cubs”.
Questions 15:Line No. 128-129: s. In this way the pups are often separated from their mothers and are never raised by their mothers. This is very loose statement.
Response 15: In order not to cause ambiguity, we have deleted these sentences as the reviewer's suggestion.
Questions 16:Line No. 133: During the period, a small amount of bamboo leaves and bamboo shoots were added to. Instead of added use the word provided, because added can be used when along with some things else, if bamboo leaves and shoots are also included. But in this case or here, it is not so. Modify the sentence ‘During the period, a small amount of bamboo leaves and shoots were provided for the juveniles gradually to switchover from milk feeding to fodder feeding.
Response 16: Considering the reviewer's suggestion, we have modified the sentence as “During the period, a small amount of bamboo leaves and shoots were provided for the juveniles gradually to switchover from milk feeding to fodder feeding.” in line 127-128 of the revised manuscript.
Questions 17:Line No: 134- : Until the age of 2, all the cubs would left their
mother and lived independently. This statement is technically and grammatically wrong. Correct it. The meaning of the sentence is that from birth until the age of 2 years, cubs independent. Is that what the authors wants to say or authors want to say that ‘at the age of 2 years onwards, all the cubs in the present study are independent of mothers’.
Response 17: Considering the reviewer's suggestion, we have modified the sentence as “At the age of 2 years onwards, all the cubs in the present study are independent of mothers.” in line 128-129 of the revised manuscript.
Questions 18:Line No. 230-231: Figure 1a show the top 20 microbes by relative abundance at the phylum level of the gut microbiome of adult captive giant pandas. The authors need not to write an exclusive sentence to cite a figure or table, instead, the figure/table can be cited in the first sentence of results. For example, in the first paragraph of results section, the second statement (line no. 231-234 ) interprets the results, whereby when figure is cited, no need for the first statement that appears in line no. 130-131. Delete the sentence.
Similar to the earlier comment, the Line No. 235-236 also meaningless to have and delete this statement.
Response 18: In order not to cause ambiguity, we have deleted these sentences as the reviewer's suggestion.
Questions 19:In line No. 238-240: Among them, the fecal microorganisms of old giant pandas are mainly Escherichia, 238 Enterobacter, Clostridium, Citrobacter, Klebsiella, Kluyvera, Lactococcus, Acinetobacter, 239 Dechloromonas, etc.
There is no need for the above statement. The old giant panda is confusing here and it is not the within the scope of this work. Delete this statement.
Response 19: In order not to cause ambiguity, we have deleted these sentences as the reviewer's suggestion.
Questions 20:Line No. 252-253: Table 2 is the rank sum test of simpson, shannon, chao1 index, showing the alpha 252 diversity results of APR group and AHR group. The authors without showing the data for the Simpson, Shanon and Chao 1, showing only the statistical test results alone. I hope they have shown this data in Table S1. I suggest the authors to show the mean±SE of the simson, shanon and chao 1 data for the two-parenting type and include the statistical test within the table. Going to the Table S1 to refer for the data is not convenient. Put them [the data and the statistical results] together in a given table and include the table as part of the manuscript and not in supplementary table.
Response 20: Thank you very much for reviewer’s suggestions. Considering the reviewer's suggestion, we have rewritten the table 2 and table S1, S2, see the revised manuscript for details.
Questions 21:Line No. 257-260: Note: Since the data of chao1 in the experimental group are normally distributed, the T-test test is needed to compare the species diversity between the PR and HR groups; however, the data of Simpson and Shannon are not normally distributed, so the Wilcox test needs to be used to compare the species diversity between the PR and HR groups.
The above Note: should be modified as follows: As data of the Chao1 are normally distributed, analysed by T-test and the data of the Simpson and Shannon are not normally distributed, tested using Wilcox Matched Pair Test. Instead of keeping this detail as a Note under Table 1, should actually be placed in the method section.
Response 21: Considering the reviewer's suggestion, we modified the sentence as “At the age of 2 years onwards, all the cubs in the present study are independent of mothers.” in line 181-183 of the revised manuscript and added this sentence in methods.
Questions 22:Line No. 275: Table 3. β diversity index of gut microbes in captive giant pandas adopting different nursing methods.
Similar to the suggestion given for Table 2, Table 3 also needs to produce the β diversity index data, besides the statistical test results.
In fact, in my earlier version of review (Question / Comment No.18), I have suggested the authors to produce the data concerned for Table 2 and Table 3, but the authors mentioned that they have included the data in Table S1, S2, and S3. As said earlier, data pertains to the Table 2 and 3 are vital and based on the data, findings are derived. In such situation, data needed to be provided
in the main table and in supplementary table.
Response 22: Thank you very much for reviewer’s suggestions. Considering the reviewer's suggestion, we have rewritten the table 3 and 4, see the revised manuscript for details.
Questions 23:Line No. 279: greater than the set value (the default value is 2) between the two groups of different species (Figure 3A). Please replace the word ‘between’ with ‘for’ and also the word ‘of’ with ‘in’. It should be as ‘ greater than the set value (the default value is 2) for the two groups giant pandas in different species (Figure 3A).
Response 23: Considering the reviewer's suggestion, we have modified the sentence as “greater than the set value (the default value is 2) for the two groups giant pandas in different species (Figure 3A)” in line 264-265 of the revised manuscript.
Questions 24:Line No. 279: greater than the set value (the default value is 2) between the two groups of different species (Figure 3A). Please replace the word ‘between’ with ‘for’ and also the word ‘of’ with ‘in’. It should be as ‘ greater than the set value (the default value is 2) for the two groups giant pandas in different species (Figure 3A).
Response 24: Considering the reviewer's suggestion, we have modified the sentence as “greater than the set value (the default value is 2) for the two groups giant pandas in different species (Figure 3A)” in line 264-265 of the revised manuscript.
Questions 25:Line No. 280: Although the differences in β diversity among the two groups were. Suggest to replace the word ‘among’ with the word ‘between’
Response 25: Considering the reviewer's suggestion, we have changed the word “among” to “between” in line 265 of the revised manuscript.
Questions 26:Authors use seem to use two different abbreviations [SCFCs & SCFAs] for a given terms i.e., short-chain fatty acids. The abbreviation SCFAs has been introduced for short-chain fatty acids in line no 432 and been used in Line No. 434 and 438. Similarly, SCFCs abbreviation has been introduced in Line No. 60 and has been used 471; for the same term, short-chain fatty acids. Authors are advised the confirm the same and be consistent.
Response 26: Sorry I didn't double check, we have changed the abbreviation of short chain fatty acids to “SCFAs” as your suggestion.
This manuscript is a resubmission of an earlier submission. The following is a list of the peer review reports and author responses from that submission.
Round 1
Reviewer 1 Report
The manuscript deals with an interesting and essential aspects for ex-situ management of a Vulnerable species [IUCN]. The cutting-edge research work has many new and novel findings using advanced tools like metagenomics and liquid chromatography mass spectrometer (LC-MS) technology, which are rarely used in wildlife studies. The method used and tools used for analyses of the data are very appropriate and update. Nevertheless, the article is suffering extensively with style of writing especially the language part including punctuations. The following are the comments and suggestions and if authors take care these, the article will contribute a lot for the management and conservation of Giant Panda in the ex-situ and in-situ.
Line No. 13-15: In the process of ex situ conservation, captive red pandas face a variety of unfavorable environmental impacts such as premature weaning, frequent maternal separation, parenting by non-parents, noise from tourists, and frequent replacement of animal houses in the early stages of life, which will cause psychological stress and stress to the captive individual.
Firstly, when the first sentence starts with In ex-situ conservation, the term captive is not an essential again the same sentence.
Secondly, as the environmental impacts such as premature weaning, frequent maternal separation, parenting by non-parents, noise from tourists, and frequent replacement of animal houses are all by and large pertain to young red pandas, why mention at first red panda (Line No. 13) and later at the end of the sentence (in Line No. 15 & 16) mention about ‘in the young stages of life.
Thirdly, in Line No. 16: which will cause ……………… to the captive individual. If these are not established facts, I suggest the authors to use ‘may cause’ instead of ‘will cause’
Finally, in the Line No. 16: psychological stress and stress to the captive individual. Use of the terms ‘psychological stress and stress’ may cause confusion, if I guess the authors are trying to say psychological stress and physiological stress.
If what I listed above is agreeable to the authors, I would suggest to modify the sentence as follows.
In the process of ex-situ conservation, young red pandas face a variety of unfavorable environmental impacts such as premature weaning, frequent maternal separation, parenting by non-parents, noise from tourists and frequent replacement of animal houses, which may cause psychological and physiological stress.
Line No. 19: So, is the stress caused by the unfavorable parenting environment in the early life of captive giant pandas related to the health of the gut?
What is the meaning of health of the gut? does it means the micro-biota of the gut. And if so, I see a contradiction between Line No. 18 and 19. Line No. 18, says stress can increase the permeability of the gut, resulting in long-term changes in the diversity and abundance of gut microbes, which means stress, an independent factor, changes the gut micro-biota, a dependent factor. However, the Line No. 19: looks like stress is dependent on the health of the gut. Could it be clarified?
Line No. 23: the traditional parenting mode. The term traditional parenting mode is not clear, as it is introduced all of a sudden.
Line No. 26: Having said that dysbiosis in the host gut microbiota is a key determinant of abnormal behavior in stress model individuals, instead of saying ‘Will this lead to stress and intestinal flora imbalance in captive giant pandas?’ in line No. 29., modifying the same into ‘Will this leads to imbalance in intestinal flora and stress in captive giant panda?’ become much more appropriate, because dysbiosis is the first order consequence of changes in early parenting environment and it is also an independent factor that cause stress and stress is the subsequent consequence, and a dependent on the gut microbiota of host.
Line No. 70: unfavorable environments such as premature weaning, separation of mother and child…..
In the above sentence, when weaning is nothing but separating the mother and young, where is the need for separation of mother and child’. Delete this, but in case if the authors mean to say temporary or short-term separation, mention it explicitly.
Line No.: 73: This can cause stress to the mother during the nursing period [7], resulting in less opportunities for learning and exercise when rearing the cubs. At the same time, it also affects the expression of social behavior of giant panda cubs to a certain extent [8]. I suggest the authors to modify the above sentence into single sentence as shown below.
This causes stress to the mothers during the nursing period [7], resulting in less opportunities for learning and exercise when rearing the cubs and also affects the social behavior of giant panda cubs to a certain extent [8].
Line No. 75 - 77: Our previous research also found that the parenting experience during the nursing period 76 (exclusive breastfeeding parenting vs. artificially assisted parenting)… The sentence is not clear. Please re-write to understand it clearly.
Line No. 78: I suggest the authors to delete the word ‘natural’
Line No. 86-89: Therefore, the purpose of this study is to use a combination of fecal metagenomics and liquid chromatography mass spectrometer (LC-MS) technology to study the effect of different parenting patterns on the structure, diversity and metabolites of the intestinal microbial community of captive giant pandas. The sentence is giving subject importance for the tool used rather than the effect of different parenting patterns on the structure, diversity and stress level. Further, Line No. 88-89: The statement ‘different parenting patterns on the structure, diversity and metabolites of the intestinal microbial community of captive giant pandas’ appears like that the study look at the metabolite of the intestinal microbial community that lives in the captive giant panda and not the metabolites of the giant panda.
I suggest the above sentence to be modified as ‘Therefore the present study evaluated the effect of different parenting pattern on the structure, and diversity of the intestinal microbial community and metabolites of captive giant pandas’ non-invasively using fecal metagenomics and liquid chromatography mass spectrometer (LC-MS) technology.
Line No. 90: Traditional parenting mode. As mentioned early in my comments, this term needs to be introduced at the introduction and authors can mention that more details about the same is given in method section. The reason for introduction the term at the introduction is needed because it is a criterion up on which the entire study is designed and hence an outline is needed at the introduction itself and the same can be placed before the Line No. 86.
Line No. 89-92: In order to evaluate the possible adverse effects of the traditional parenting mode on the intestinal health of captive giant pandas in the early life, it can provide an important scientific basis for improving the welfare level of captive giant pandas. The sentence should be rewritten as follows: Evaluating the possible adverse effects of the traditional parenting mode on the intestinal health of young captive giant pandas can provide an important scientific basis for improving the welfare level of captive giant pandas. This sentence should be placed before the Line No. 86-89. To have flow, the introduction should end with Line No. 86-89 with the modifications suggested above.
Table 1. Experimental Grouping. The table title should provide all the details for the reader to understand it clearly and the title should be stand-alone.
Expansions of the abbreviations are not described clearly for the readers to understand them easily. For example, the expansion of the Young-PR given as Young: Juvenile panda; PR: Infants are adopting exclusive breastfeeding nursing method. In this the letters PR stands for what is not clearly described or expanded. I hope the PR stands for Parent-Raised and similarly HR stands for Hand-Raised. If what suspect is right, I suggest the authors to include the details into the expansion, so that it is easier for anyone to understand while reading them in the text. The Group category column could also be simplified into Juvenile Parent-Raised (JPR), Juvenile Hand-Raised (JHR) and similarly, Adult Parent-Raised (AHR) and Adult Hand-Raised (AHR). These expanded form with abbreviations could be included into the Group column and a note for each abbreviation given with star sign, for example abbreviation for YPR may be given with one star as ‘YPR*’ and the single * indicates juveniles in this group are ‘exclusively with breastfeeding and nursing by Parents alone’ can be given at the bottom of the table. Same way, authors could follow for the rest of abbreviations as YHR**, APR*** and AHR****.
Line No. 109-111: According to the conventional management experience of captive giant pandas, the nursing period of captive giant pandas is 0-2 years old, and the sub-adult period of captive giant pandas is 2-7 years old.
In the above authors are (i) mixing up age-class (sub-adult) and nursing period (0-2 years old) , (ii) also not sighted any standard reference, (iii) not mentioned about all the age-classes in general (adult, juvenile, cub). (iv) I also suspect whether sub-adults age go up 7 years. Therefore, I suggest let the authors first describe the various age- classes like, young (cub), juvenile, sub-adult and adult giving age details for each age class and citing a standard reference at the end. And then describe about the nursing period as 0-2 years sighting a standard reference again or say based on captive management.
Line No. 111-113: during the nursing period, the traditional parenting methods of captive giant pandas are divided into exclusive breastfeeding nursing method and artificial assisted nursing method. I suggest the above to be modified as follows.
In the present study, the parenting methods of captive giant pandas are categorized into exclusive breastfeeding nursing method by parents (PR) and artificial assisted nursing method (HR).
Line No. 121: the artificial assisted nursing method is mainly aimed at the mother animals with.
In this delete the word animals and add ‘s’ to mother
Line No. 139: centrifuge at 12,000 g. Is this 12000 g or rpm?
Line No. 146: For each sample, Replace the word ‘For’ with ‘From’
Line No. 159: the final processing result obtains the valid sequence. It should be as ‘finally processed the results obtaining valid sequence’
Line No. 169-171: Break this as a new sentence and modify it as suggested. Analyses such as diversity analysis (α diversity 169 and β Diversity analysis), difference analysis (species Venn diagram, linear discriminant analysis (LDA), effect size analysis, partial least squares discriminant analysis (PLS-DA) and random forest analysis were carried out using software ????? version ???.
Line No. 174: Remove the word ‘After’
Line No. 177-179: In these line steps are described in present tense for example centrifuge at 14000g, take the supernatant, dry it in vacuo …… take supernatant sample for analysis. As all these steps are already completed and authors are reporting the steps completed, they have to be past tense.
Line No. 194: please change The detailed data information of the 24 samples sees Table S1 as ‘The detailed data information of the 24 samples are shown Table S1’
Line No. 195. Replace the word ‘showed’ as ‘show’
Line No. 196-200: From the results in Figure 1A and B, it could be seen that the intestinal flora of each group of giant pandas, both juvenile and adult, was dominated by Proteobacteria, Firmicutes, Actinobacteria, Uroviricota, Bacteroidetes and Basidiomycota. Among them, Proteobacteria and Firmicutes were the dominant bacterial phyla, accounting for more than 75% of the total microbial community. Modify the same as ‘From the results in Figure 1A and B …… Proteobacteria, Firmicutes …… Bacteroidetes and Basidiomycota, with first two phyla accounting for 75% of the total microbial community.
Figure 1: Does not show the legend of various colours representing what taxon.
Line No. 211: ………………..group increased. Use the terms such as higher or lower. For example, in Young-HR compared to Yong-PR. Similarly, all comparisons should be made for each phylum or all phyla between the two parenting types, instead saying a phylum increased in given parenting type, while some other phylum decreased compared with another parenting type. Because, the present study deals with variables with categorical data (YHR versus YPR) and not with continuous data/variable. If it is continuous data / variable for example, age of the red panda, you can use a given phylum or set of phyla increased with age and another set of phyla decreased.
Table citation in the text should be in order, Table 3 cited in the text, and the Table 2 is sighted latter.
I do not see the data pertain to Table 2 (α diversity index of gut microbes) or Table 3 (β diversity index of gut microbes). What has been shown in these two tables are statistical results, but their data neither shown in main file not in supplementary files.
232: Delete the words ‘in the tract microorganisms’
Line No. 226-236 runs without break with ‘coma or semicolon’. Advised to break into many sentences.
Line No. 332-341: Should go under methodology.
Line No. 363-365: These should also be moved to methodology.
Line No. 405: Instead of saying ‘fed with artificially assisted feeding’ saying ‘artificial food with assisted feeding’ because the food fed here is artificial and that pandas fed on them with human assistance.
Line No. 422: Delete the citation of statistical test i.e. (non-parametric factorial kruskal-wallis sum-rank 422 test, LDA > 3).
Line No. 421-425: There is some contradiction between the two points mentioned below.
(i) Relative abundance of Clostridium in the Old-PR group was higher than that in the Old-HR group.
(ii) This seemed to indicate that the maternal separation in the early parenting stage had caused stress to the captive giant pandas in adulthood, as previous studies had demonstrated that stress caused an increase in the host microorganism (Clostridium) [22].
As per the description/justification given in the point ii, maternal separation in the early parenting stage (that takes place mostly in Old-HR group) resulted in stress to captive giant pandas(22) and stress increases the host microorganism (Clostridium) means, it is the Old-HR group should have had higher level of Clostridium, but in the present findings, it is the opposite i.e., the Old-PR group had Clostridium than that of Old-HR. Therefore, comparison of present findings with earlier is contradicting. Please check whether your interpretation of the earlier study is wrong or the present result is contradicting the earlier observed trend.
Line No. 424-425: Ss previous studies had 424 demonstrated that stress caused an increase in the host microorganism (Clostridium) [22]. The reference cited in the current context actually deals about ‘Wheat or rye supplemented diets do not affect faecal mucus concentration or the adhesion of probiotic micro-organisms to faecal mucus. Lett Appl Microbiol. 2010, 31: 30-33’. Going by the title of the article it looks inappropriate to refer in the context of the sentence, as it does not deal about the stress component. Therefore, I suggest the authors to confirm the same and if so, sight relevant one.
Line No. 443-446: The results of this experiment are a good combination of the two conclusions, revealing the psychological mechanism of early parenting environment affecting the decline of natural mating behavior of captive giant pandas from the perspective of gut microbes.
Any study should make conclusions only based on aspects dealt, data collected and obtained results of the investigation and not based on the aspects not dealt by the study or without having any data or findings. I suggest the authors to focus the conclusions with the aspect dealt and results obtained. As there is separate section on conclusion in line no 448-453, I feel the sentence in 443-446 could be deleted, as the study does not have data on mating behaviour of captive giant panda.
The manuscript deals with interesting findings and useful for conservation. However, besides the above there are many more places in the text, where the flow is missing. Authors are suggested to get the manuscript edited by some one who is good English language.
Author Response
We thank you very much for giving us an opportunity to revise our manuscript, we appreciate editor and reviewers very much for their positive and constructive comments and suggestions on our manuscript entitled “Effects of early nursery environment on the gut health of captive giant pandas”. (ID: animals- 1841981).
Specific responds to the reviewer’s comments:
Questions 1:Line No. 13-15: In the process of ex situ conservation, captive red pandas face a variety of unfavorable environmental impacts such as premature weaning, frequent maternal separation, parenting by non-parents, noise from tourists, and frequent replacement of animal houses in the early stages of life, which will cause psychological stress and stress to the captive individual.
Firstly, when the first sentence starts with In ex-situ conservation, the term captive is not an essential again the same sentence.
Secondly, as the environmental impacts such as premature weaning, frequent maternal separation, parenting by non-parents, noise from tourists, and frequent replacement of animal houses are all by and large pertain to young red pandas, why mention at first red panda (Line No. 13) and later at the end of the sentence (in Line No. 15 & 16) mention about ‘in the young stages of life.
Thirdly, in Line No. 16: which will cause ……………… to the captive individual. If these are not established facts, I suggest the authors to use ‘may cause’ instead of ‘will cause’
Finally, in the Line No. 16: psychological stress and stress to the captive individual. Use of the terms ‘psychological stress and stress’ may cause confusion, if I guess the authors are trying to say psychological stress and physiological stress.
If what I listed above is agreeable to the authors, I would suggest to modify the sentence as follows.
In the process of ex-situ conservation, young red pandas face a variety of unfavorable environmental impacts such as premature weaning, frequent maternal separation, parenting by non-parents, noise from tourists and frequent replacement of animal houses, which may cause psychological and physiological stress.
Response 1: Considering the reviewer's suggestion, we have modify the sentence as “In the process of ex-situ conservation, young red pandas face a variety of unfavorable environmental impacts such as premature weaning, frequent maternal separation, parenting by non-parents, noise from tourists and frequent replacement of animal houses, which may cause psychological and physiological stress”.
Questions 2:Line No. 19: So, is the stress caused by the unfavorable parenting environment in the early life of captive giant pandas related to the health of the gut?
What is the meaning of health of the gut? does it means the micro-biota of the gut. And if so, I see a contradiction between Line No. 18 and 19. Line No. 18, says stress can increase the permeability of the gut, resulting in long-term changes in the diversity and abundance of gut microbes, which means stress, an independent factor, changes the gut micro-biota, a dependent factor. However, the Line No. 19: looks like stress is dependent on the health of the gut. Could it be clarified?
Response 2: Thank you very much for reviewer’s suggestions, as your suggestion,we have rewritten these sentences as “Recent observations had revealed an association between stress and alterations of the intestinal microbiota [9]. These studies suggested that specific gut bacteria could restrain the activation of the HPA axis, and affected social behaviors [10]” And added two references to prove our point.
Questions 3:Line No. 23: the traditional parenting mode. The term traditional parenting mode is not clear, as it is introduced all of a sudden.
Response 3: Considering the reviewer's suggestion, we have modified this sentence as “Since the newborn cubs of giant pandas are very weak, in order to improve the survival rate of giant panda cubs, the traditional parenting mode usually adopts artificial assistance, that is, manual intervention is adopted in the early parenting stage. The advantage is that it can ensure the safety of the cubs and improve the survival rate of the cubs.” to a detailed introduction to traditional parenting mode.
Questions 4:Line No. 26: Having said that dysbiosis in the host gut microbiota is a key determinant of abnormal behavior in stress model individuals, instead of saying ‘Will this lead to stress and intestinal flora imbalance in captive giant pandas?’ in line No. 29., modifying the same into ‘Will this leads to imbalance in intestinal flora and stress in captive giant panda?’ become much more appropriate, because dysbiosis is the first order consequence of changes in early parenting environment and it is also an independent factor that cause stress and stress is the subsequent consequence, and a dependent on the gut microbiota of host.
Response 4: Considering the reviewer's suggestion, we have modified this sentence as “Will this leads to imbalance in intestinal flora and stress in captive giant panda?”.
Questions 5:Line No. 70: unfavorable environments such as premature weaning, separation of mother and child…..
In the above sentence, when weaning is nothing but separating the mother and young, where is the need for separation of mother and child’. Delete this, but in case if the authors mean to say temporary or short-term separation, mention it explicitly.
Response 5: Considering the reviewer's suggestion, we have deleted this sentence.
Questions 6:Line No.: 73: This can cause stress to the mother during the nursing period [7], resulting in less opportunities for learning and exercise when rearing the cubs. At the same time, it also affects the expression of social behavior of giant panda cubs to a certain extent [8]. I suggest the authors to modify the above sentence into single sentence as shown below.
This causes stress to the mothers during the nursing period [7], resulting in less opportunities for learning and exercise when rearing the cubs and also affects the social behavior of giant panda cubs to a certain extent [8].
Response 6: Considering the reviewer's suggestion, we have modified this sentence as ” This causes stress to the mothers during the nursing period [7], resulting in less opportunities for learning and exercise when rearing the cubs and also affects the social behavior of giant panda cubs to a certain extent [8]”.
Questions 7:Line No. 75 - 77: Our previous research also found that the parenting experience during the nursing period 76 (exclusive breastfeeding parenting vs. artificially assisted parenting)… The sentence is not clear. Please re-write to understand it clearly.
Line No. 78: I suggest the authors to delete the word ‘natural’
Response 7: Considering the reviewer's suggestion, we have deleted this sentence and word.
Questions 8:Line No. 86-89: Therefore, the purpose of this study is to use a combination of fecal metagenomics and liquid chromatography mass spectrometer (LC-MS) technology to study the effect of different parenting patterns on the structure, diversity and metabolites of the intestinal microbial community of captive giant pandas. The sentence is giving subject importance for the tool used rather than the effect of different parenting patterns on the structure, diversity and stress level. Further, Line No. 88-89: The statement ‘different parenting patterns on the structure, diversity and metabolites of the intestinal microbial community of captive giant pandas’ appears like that the study look at the metabolite of the intestinal microbial community that lives in the captive giant panda and not the metabolites of the giant panda.
I suggest the above sentence to be modified as ‘.
Response 8: Considering the reviewer's suggestion, we have modified this sentence as ” Therefore the present study evaluated the effect of different parenting pattern on the structure, and diversity of the intestinal microbial community and metabolites of captive giant pandas’ non-invasively using fecal metagenomics and liquid chromatography mass spectrometer (LC-MS) technology”.
Questions 9:Line No. 90: Traditional parenting mode. As mentioned early in my comments, this term needs to be introduced at the introduction and authors can mention that more details about the same is given in method section. The reason for introduction the term at the introduction is needed because it is a criterion up on which the entire study is designed and hence an outline is needed at the introduction itself and the same can be placed before the Line No. 86.
Response 9: Considering the reviewer's suggestion, we have added some sentences to introduce the traditional parenting mode in the introduction and methods.
Questions 10:Line No. 89-92: In order to evaluate the possible adverse effects of the traditional parenting mode on the intestinal health of captive giant pandas in the early life, it can provide an important scientific basis for improving the welfare level of captive giant pandas. The sentence should be rewritten as follows: Evaluating the possible adverse effects of the traditional parenting mode on the intestinal health of young captive giant pandas can provide an important scientific basis for improving the welfare level of captive giant pandas. This sentence should be placed before the Line No. 86-89. To have flow, the introduction should end with Line No. 86-89 with the modifications suggested above.
Response 10: Considering the reviewer's suggestion, we have modified this sentence as ” Evaluating the possible adverse effects of the traditional parenting mode on the intestinal health of young captive giant pandas can provide an important scientific basis for improving the welfare level of captive giant pandas.” And placed before the Line No. 86-89.
Questions 11:Table 1. Experimental Grouping. The table title should provide all the details for the reader to understand it clearly and the title should be stand-alone.
Expansions of the abbreviations are not described clearly for the readers to understand them easily. For example, the expansion of the Young-PR given as Young: Juvenile panda; PR: Infants are adopting exclusive breastfeeding nursing method. In this the letters PR stands for what is not clearly described or expanded. I hope the PR stands for Parent-Raised and similarly HR stands for Hand-Raised. If what suspect is right, I suggest the authors to include the details into the expansion, so that it is easier for anyone to understand while reading them in the text. The Group category column could also be simplified into Juvenile Parent-Raised (JPR), Juvenile Hand-Raised (JHR) and similarly, Adult Parent-Raised (AHR) and Adult Hand-Raised (AHR). These expanded form with abbreviations could be included into the Group column and a note for each abbreviation given with star sign, for example abbreviation for YPR may be given with one star as ‘YPR*’ and the single * indicates juveniles in this group are ‘exclusively with breastfeeding and nursing by Parents alone’ can be given at the bottom of the table. Same way, authors could follow for the rest of abbreviations as YHR**, APR*** and AHR****.
Response 11: We have modified this sentence and added some information as the reviewer's suggestion.
Questions 12:Line No. 109-111: According to the conventional management experience of captive giant pandas, the nursing period of captive giant pandas is 0-2 years old, and the sub-adult period of captive giant pandas is 2-7 years old.
In the above authors are (i) mixing up age-class (sub-adult) and nursing period (0-2 years old) , (ii) also not sighted any standard reference, (iii) not mentioned about all the age-classes in general (adult, juvenile, cub). (iv) I also suspect whether sub-adults age go up 7 years. Therefore, I suggest let the authors first describe the various age- classes like, young (cub), juvenile, sub-adult and adult giving age details for each age class and citing a standard reference at the end. And then describe about the nursing period as 0-2 years sighting a standard reference again or say based on captive management.
Response 12: We have modified this sentence and added some management information and reference to prove it as the reviewer's suggestion.
Questions 13:Line No. 111-113: during the nursing period, the traditional parenting methods of captive giant pandas are divided into exclusive breastfeeding nursing method and artificial assisted nursing method. I suggest the above to be modified as follows.
In the present study, the parenting methods of captive giant pandas are categorized into exclusive breastfeeding nursing method by parents (PR) and artificial assisted nursing method (HR).
Response 13: Considering the reviewer's suggestion, we have modified this sentence as ” In the present study, the parenting methods of captive giant pandas are categorized into exclusive breastfeeding nursing method by parents (PR) and artificial assisted nursing method (HR)”
Questions 14:Line No. 121: the artificial assisted nursing method is mainly aimed at the mother animals with.
In this delete the word animals and add ‘s’ to mother
Line No. 139: centrifuge at 12,000 g. Is this 12000 g or rpm?
Line No. 146: For each sample, Replace the word ‘For’ with ‘From’
Line No. 159: the final processing result obtains the valid sequence. It should be as ‘finally processed the results obtaining valid sequence’
Line No. 169-171: Break this as a new sentence and modify it as suggested. Analyses such as diversity analysis (α diversity 169 and β Diversity analysis), difference analysis (species Venn diagram, linear discriminant analysis (LDA), effect size analysis, partial least squares discriminant analysis (PLS-DA) and random forest analysis were carried out using software ????? version ???.
Line No. 174: Remove the word ‘After’
Line No. 177-179: In these line steps are described in present tense for example centrifuge at 14000g, take the supernatant, dry it in vacuo …… take supernatant sample for analysis. As all these steps are already completed and authors are reporting the steps completed, they have to be past tense.
Response 14: Considering the reviewer's suggestion, we have modified these sentences and words and checked the speed of centrifuge.
Questions 15:Line No. 194: please change The detailed data information of the 24 samples sees Table S1 as ‘The detailed data information of the 24 samples are shown Table S1’
Line No. 195. Replace the word ‘showed’ as ‘show’
Line No. 196-200: From the results in Figure 1A and B, it could be seen that the intestinal flora of each group of giant pandas, both juvenile and adult, was dominated by Proteobacteria, Firmicutes, Actinobacteria, Uroviricota, Bacteroidetes and Basidiomycota. Among them, Proteobacteria and Firmicutes were the dominant bacterial phyla, accounting for more than 75% of the total microbial community. Modify the same as ‘From the results in Figure 1A and B …… Proteobacteria, Firmicutes …… Bacteroidetes and Basidiomycota, with first two phyla accounting for 75% of the total microbial community.
Response 15: Considering the reviewer's suggestion, we have modified these sentences and words.
Questions 16:Figure 1: Does not show the legend of various colours representing what taxon
Response 16: Considering the reviewer's suggestion, we have rewritten the figure 1.
Questions 17:Line No. 211: ………………..group increased. Use the terms such as higher or lower. For example, in Young-HR compared to Yong-PR. Similarly, all comparisons should be made for each phylum or all phyla between the two parenting types, instead saying a phylum increased in given parenting type, while some other phylum decreased compared with another parenting type. Because, the present study deals with variables with categorical data (YHR versus YPR) and not with continuous data/variable. If it is continuous data / variable for example, age of the red panda, you can use a given phylum or set of phyla increased with age and another set of phyla decreased.
Response 17: Considering the reviewer's suggestion, we have modified these words.
Questions 18:Table citation in the text should be in order, Table 3 cited in the text, and the Table 2 is sighted latter.
I do not see the data pertain to Table 2 (α diversity index of gut microbes) or Table 3 (β diversity index of gut microbes). What has been shown in these two tables are statistical results, but their data neither shown in main file not in supplementary files.
Response 18: Considering the reviewer's suggestion, we have added the supplement table S1, 2 and 3.
Questions 19:232: Delete the words ‘in the tract microorganisms’
Line No. 226-236 runs without break with ‘coma or semicolon’. Advised to break into many sentences.
Line No. 332-341: Should go under methodology.
Line No. 363-365: These should also be moved to methodology.
Line No. 405: Instead of saying ‘fed with artificially assisted feeding’ saying ‘artificial food with assisted feeding’ because the food fed here is artificial and that pandas fed on them with human assistance.
Line No. 422: Delete the citation of statistical test i.e. (non-parametric factorial kruskal-wallis sum-rank 422 test, LDA > 3).
Response 19: Considering the reviewer's suggestion, we have modified these sentences and words.
Questions 20:Line No. 421-425: There is some contradiction between the two points mentioned below.
(i) Relative abundance of Clostridium in the Old-PR group was higher than that in the Old-HR group.
(ii) This seemed to indicate that the maternal separation in the early parenting stage had caused stress to the captive giant pandas in adulthood, as previous studies had demonstrated that stress caused an increase in the host microorganism (Clostridium) [22].
As per the description/justification given in the point ii, maternal separation in the early parenting stage (that takes place mostly in Old-HR group) resulted in stress to captive giant pandas(22) and stress increases the host microorganism (Clostridium) means, it is the Old-HR group should have had higher level of Clostridium, but in the present findings, it is the opposite i.e., the Old-PR group had Clostridium than that of Old-HR. Therefore, comparison of present findings with earlier is contradicting. Please check whether your interpretation of the earlier study is wrong or the present result is contradicting the earlier observed trend.
Response 20: I am sorry that I did not carefully review the discussion part of the article, you are very correct; we have re-analyzed the obtained data and discussed the results in the revised manuscript, and added metabolites and microbial correlations analysis, which can more favorably prove our conclusion.
Questions 21:Line No. 443-446: The results of this experiment are a good combination of the two conclusions, revealing the psychological mechanism of early parenting environment affecting the decline of natural mating behavior of captive giant pandas from the perspective of gut microbes.
Any study should make conclusions only based on aspects dealt, data collected and obtained results of the investigation and not based on the aspects not dealt by the study or without having any data or findings. I suggest the authors to focus the conclusions with the aspect dealt and results obtained. As there is separate section on conclusion in line no 448-453, I feel the sentence in 443-446 could be deleted, as the study does not have data on mating behaviour of captive giant panda.
Response 21: In accordance with the reviewer's suggestion, we have removed this part of the content that is likely to cause ambiguity to readers.

Reviewer 2 Report
Effects of early nursery environment on the gut health of captive giant pandas
The article investigates the consequences of maternal deprivation in early life, prior to weaning, on the gut flora and metabolites of captive giant panda. The work highlights to potential problems that can arise from these common breeding practices. The authors assess differences in the gut flora, assessed by metagenomics, to show that the composition of the gut flora did not differ between the two groups but the relative abundance of certain types of microbes did. Further they assessed metabolites, using LC-MS methods to related these microbial differences to the presence of metabolites. The authors, then draw on the literature and their own previous work to relate these shifts in gut flora and metabolites to behavioural and reproductive problems in adult panda. The paper is interesting and will help to improve captive breeding protocols for this species, which is pertinent, as high production of maladapted individuals is not beneficial to breeding programmes in the long term.
The authors present a straightforward, testable hypothesis for which the methodologies applied appear appropriate. It is clear and concisely written. However, the English language could be improved as there were several very long and complex sentences, that were difficult to understand. I have not, as directed by the journal review guidelines, provided detailed comments on this but addressing this will improve the clarity of the article. Also, the fonts in several of the figures are very small and difficult to read.
I only have a few comments:
The article is focused on giant panda, but I think these issues are likely to be relevant a range of species. Certainly, the gut flora of captive and wild animals often differs, and while adjustment is possible I wondered if the authors could broaden the context of their work in the discussion, to highlight other situations where similar captive breeding practices may lead to similar issues.
Could the authors comment on the proportion of juveniles consuming bamboo in each group and if this could lead to differences in the microbial floral of individuals that are associated with a shift to bamboo consumption rather than separation? Could this have influenced the observed differences? Also, how many of the animals used are twins? If any, could there be any impacts of related individuals influencing the results, i.e. similar individuals possessing similar microbiome because of shared traits unrelated to the treatment of separation?
Could the authors please clarify if negative controls were collected and processed as part of the metagenomics methodology to test for any potential contamination at any stage of the process? And please include the results of these tests. This is important because microbial contamination can easily occur and this shows it has not impacted results.
Specific comments:
Line 50: how were samples purified to the specified length? Can you clarify if samples were size selected to 200-500bp or if fragments under 200bp were removed?
Line 160; How were the metagenomic contigs created?
Line 390: add the reference at the end of this sentence.
Throughout: Please define the PR and HR in the main text as well as the table legend.
Table 1: It would be helpful to not have the Groups centred. It makes it more difficult to see which individuals are associated with which groups and how many individuals in each of the juvenile groups
Figure 1: It would be helpful if the identification of the microbes for the different colours is included in either the figure or the legend
Figure 4: I’m not sure part c of this figure is very informative for the amount of space it takes up, it would be better if b was increased in size, as it is very difficult to read.
Figure 5: again, the font here is very small, making it difficult to read.
Author Response
We thank you very much for giving us an opportunity to revise our manuscript, we appreciate editor and reviewers very much for their positive and constructive comments and suggestions on our manuscript entitled “Effects of early nursery environment on the gut health of captive giant pandas”. (ID: animals- 1841981).
Specific responds to the reviewer’s comments:
Questions 1:The authors present a straightforward, testable hypothesis for which the methodologies applied appear appropriate. It is clear and concisely written. However, the English language could be improved as there were several very long and complex sentences, that were difficult to understand. I have not, as directed by the journal review guidelines, provided detailed comments on this but addressing this will improve the clarity of the article. Also, the fonts in several of the figures are very small and difficult to read.
Response 1: Thank you very much for reviewer’s suggestions. We have tried our best to revise our manuscript. We asked a friend who have stayed in America for more than four years to revise the manuscript.
Questions 2:The article is focused on giant panda, but I think these issues are likely to be relevant a range of species. Certainly, the gut flora of captive and wild animals often differs, and while adjustment is possible I wondered if the authors could broaden the context of their work in the discussion, to highlight other situations where similar captive breeding practices may lead to similar issues.
Response 2: Considering the reviewer's suggestion, we have added the sentence as “Because the study of the intestinal flora of siberian tigers, which are also endangered wild animals, found that the living environment could significantly change the alpha diversity of the intestinal flora of siberian tigers, resulting in significant differences in gut microbiota composition and function between captive and wild siberian tigers populations [20]. Likewise, the relative abundances of the gut microbiota between captive and wild forest musk deer were significantly different [21]”, and two references to prove my opinion.
Questions 3:Could the authors comment on the proportion of juveniles consuming bamboo in each group and if this could lead to differences in the microbial floral of individuals that are associated with a shift to bamboo consumption rather than separation? Could this have influenced the observed differences? Also, how many of the animals used are twins? If any, could there be any impacts of related individuals influencing the results, i.e. similar individuals possessing similar microbiome because of shared traits unrelated to the treatment of separation?
Response 3: Considering the reviewer's suggestion, we have added the sentence as “All twins in the APR group were fed by mixed feeding of artificial formula milk and breast milk, and all singletons were exclusively breastfed. All the nursing stage in the AHR group was mixed feeding with artificial formula milk and breast milk.” and added some information of management methods of giant pandas at different growth stages in the methods.
Questions 4:Could the authors please clarify if negative controls were collected and processed as part of the metagenomics methodology to test for any potential contamination at any stage of the process? And please include the results of these tests. This is important because microbial contamination can easily occur and this shows it has not impacted results.
Response 4: Sorry, since the experiment ended last year, the negative control cannot be supplemented according to your request, but we can guarantee that the collection of fecal samples is carried out in accordance with strict procedures to minimize the contamination of the samples by the external environment.
Questions 5:Line 50: how were samples purified to the specified length? Can you clarify if samples were size selected to 200-500bp or if fragments under 200bp were removed?
Line 160; How were the metagenomic contigs created?
Line 390: add the reference at the end of this sentence.
Response 5: We have added some sentences in the methods as the reviewer's suggestions.
Questions 6:Throughout: Please define the PR and HR in the main text as well as the table legend.
Table 1: It would be helpful to not have the Groups centred. It makes it more difficult to see which individuals are associated with which groups and how many individuals in each of the juvenile groups
Response 6: Considering the reviewer's suggestion, we have rewritten the table 1.
Questions 7:Figure 1: It would be helpful if the identification of the microbes for the different colours is included in either the figure or the legend
Figure 4: I’m not sure part c of this figure is very informative for the amount of space it takes up, it would be better if b was increased in size, as it is very difficult to read.
Figure 5: again, the font here is very small, making it difficult to read.
Response 7: Considering the reviewer's suggestion, we have rewritten the all the figures.
